# Computed Tomography Anatomy of the Juvenile Cory’s Shearwater (*Calonectris borealis)* Normal Nasal Cavity

**DOI:** 10.3390/ani14203015

**Published:** 2024-10-18

**Authors:** Alejandro Morales-Espino, Marcos Fumero-Hernández, Francisco Suárez-Cabrera, Mario Encinoso, Magnolia María Conde-Felipe, Jose Raduan Jaber

**Affiliations:** 1IVC Evidensia Los Tarahales, 35013 Las Palmas, Gran Canaria, Spain; alejandro.morales108alu@gmail.com; 2Department of Morphology, Faculty of Veterinary Medicine, University of Las Palmas de Gran Canaria, Trasmontaña, Arucas, 35413 Las Palmas, Gran Canaria, Spain; marcos.vet@outlook.es (M.F.-H.); ecovetcanarias@yahoo.es (F.S.-C.); magnolia.conde@ulpgc.es (M.M.C.-F.); 3Hospital Clínico Veterinario, Facultad de Veterinaria, Universidad de Las Palmas de Gran Canaria, Trasmontaña, Arucas, 35413 Las Palmas, Gran Canaria, Spain; 4VETFUN, Educational Innovation Group, University of Las Palmas de Gran Canaria, Trasmontaña, Arucas, 35413 Las Palmas, Gran Canaria, Spain

**Keywords:** cross-sectional anatomy, computed tomography, nasal cavity, paranasal sinuses, cranial region, Cory’s shearwater

## Abstract

Modern imaging diagnostic techniques, including computed tomography (CT) with anatomical sections, were used to evaluate the nasal cavity of Cory’s Shearwater (*Calonectris borealis*). These techniques effectively delineated the primary formations within the nasal cavity and its related formations. This investigation pioneered the use of anatomical sections and computerized tomography to describe the nasal cavity in Cory’s Shearwater.

## 1. Introduction

The nasal cavity represents the first segment of the respiratory system in birds. It begins at the nares, the openings responsible for allowing air to enter this structure. Its termination occurs in the oropharynx, where respiratory and digestive functions are connected [1,2,3]. The main function of the upper respiratory tract is to filter, warm, and moisten the inhaled air. In terms of internal anatomy, birds have a series of complex structures called nasal conchae, which are delicate bony formations arranged in a rostrocaudal, rather than dorsoventral, sequence [1,2,3,4,5]. These conchae create spaces between them, known as nasal meatuses, through which both inhaled and exhaled air flows during the respiratory process [1,2,3,4,5,6,7,8,9]. The caudoventral communication of the nasal cavity with the nasopharynx takes place through the choanae or nasopharyngeal openings. This connection is essential for proper air exchange between the nasal cavity and the nasopharynx [1,2,3,4,5,6,7,8,9,10,11,12]. For an adequate understanding, it is pivotal to note that the nasal cavity is separated into two parts by the nasal septum. The floor of the cavity is substantially supported anteriorly by the palatine processes of the maxilla and premaxilla, but posteriorly, the only bones present are the slender palatine bone and the vomer [1,2,3,13]. As it extends towards the rostral portion, the nasal septum becomes cartilaginous, allowing greater flexibility and adaptation to the anatomy of the beak.

Beak anatomy depends on the bird’s lifestyle, revealing adaptations between different species [14,15,16,17]. *Calonectris borealis*, also known as Cory’s Shearwater, presents a hook shape beak, which is perfectly designed for its piscivorous diet, being long, slender, and slightly curved at the tip, ideal for catching and grasping slippery prey such as fish and squid [18,19,20,21,22,23].

Cory’s Shearwater is a pelagic seabird belonging to the family Procellariidae and plays a crucial role in the marine ecosystem. Nevertheless, despite its environmental importance, Cory’s Shearwater faces significant ecological challenges due to human activity and environmental pressures. Its classification as ‘Least Concern’ on the IUCN Red List of Threatened Species highlights the need for increased conservation efforts [23]. Therefore, understanding the complexities of their anatomy and physiology becomes crucial in scientific research, providing valuable insights into evolutionary adaptations and ecological resilience.

Hence, the advances brought by these imaging procedures have been essential in the investigation of this discipline. Initially, since its inception, radiology has been the tool of choice for exploring the nasal cavity in animals [4,5,6,7,8,9,10,11,12,13,24,25,26,27,28,29,30,31,32,33]. However, its application is limited by overlapping structures, making accurate visualization difficult. However, computed tomography (CT) and magnetic resonance imaging (MRI) have emerged as more precise and reliable alternatives for the study of avian anatomy and associated pathologies [5,8,9,10,11,12,13,33,34,35,36,37,38,39]. In contrast to MRI, the CT technique surpasses traditional radiology in sensitivity and accuracy, allowing for a more detailed evaluation of the nasal cavity. To date, the application of computed tomography (CT) in the assessment of the nasal cavity anatomy in birds is poorly documented in the veterinary literature. Although some reports describing avian nasal anatomy using CT have been found [8,9,33], research in this area remains limited. Furthermore, to the authors’ knowledge, no publications have been found that combine CT use with macroscopic sections to examine detailed anatomy of the nasal cavity of shearwaters. Hence, this research aims to describe the nasal cavity of Cory’s Shearwater using anatomical sections and computed tomography (CT) studies.

## 2. Materials and Methods

### 2.1. Animals

We worked with 8 juvenile Cory’s Shearwater (*Calonectris borealis*) cadavers, kindly provided by the Consejería de Área de Medio Ambiente, Clima, Energía y Conocimiento of the Cabildo Insular de Gran Canaria. These birds exhibited an average weight of 0.520 kg (with a range of 0.480 to 0.820 kg) and an average size of 52 cm (ranging from 45 to 56 cm) from beak to tail base. In addition, measurements of the skull were taken from the vertex of the beak to the occipital, with a mean of 12.48 cm. Similarly, the length of the nasal cavity was calculated, from the base of the beak to the rostral area of the orbit, with an average of 7.5 cm. The specimens had stranded as a consequence of artificial lightning, which is a documented hazard to biodiversity conservation [39,40,41]. Although most birds were collected post-mortem, those that were initially alive but succumbed due to their debilitated state were preserved rapidly by freezing for subsequent CT study. This meticulous approach facilitated precise identification and correlation with the CT images, enhancing the depth of our anatomical investigations. We also declare that no animal was euthanized or captured intentionally for scientific purposes.

### 2.2. CT Technique

For CT scan evaluation, our avian specimens were thawed at room temperature for 12 h. We conducted transverse CT scans with a 16-slice helical CT scanner (Toshiba Astelion, Canon Medical System^®^, Tokyo, Japan). The birds were placed in symmetrical dorsal recumbency on the couch with a craniocaudal entry. Using a standard protocol, we applied the following parameters: 120 kVp, 80 mA, 512 × 512 acquisition matrix, 1809 × 858 fields of view, a pitch of 0.94, and a gantry rotation of 1.5. These images had a slice thickness of 0.6 mm. To improve the identification of the anatomical formations on CT scans, we used different CT window settings by adjusting the window widths (WWs) and window levels (WLs): a bone window setting (WW = 1500; WL = 300) and a lung window setting (WW = 1400; WL = −500). No significant variations in CT density or anatomy were observed in the nasal cavity of the subjects included in this research. Additionally, we utilized the original data to generate volume-rendered reconstructed images using a standard Dicom 3D format (OsiriX MD, Geneva, Switzerland).

### 2.3. Anatomic Evaluation

We made anatomical cross-sections to help in the visualization of those formations observed in the computed tomography (CT) scans. After the CT study, the birds were successively stored in a freezer at −80 °C until completely frozen. Then, three frozen carcasses were cross-sectioned with an electric band saw to obtain anatomical sequential transverse, sagittal, and dorsal anatomical slices. Contiguous 1 cm cross-sections were made, starting at the beak and extending to the ocular orbit. These slices were cleaned with water, numbered, and photographed on both cranial and caudal surfaces. Once the sections were imaged, we selected those that matched the CT images. This process allowed us to more precisely identify the significant structures in the nasal cavity of the shearwater. To complement and verify our findings, we used specific anatomical texts and notable references previously described in these species [1,2,3,4,5,6,7,8,9,10,11,12,13,34].

## 3. Results

Here, we present an anatomical imaged view, a sagittal MPR, and a dorsal CT image, where each line and number indicates the approximate levels of the subsequent anatomical and cross-sectional CT images (Figure 1). Those sections highlight the relevant anatomical structures of the nasal cavity of Cory’s Shearwater (Figure 2, Figure 3, Figure 4, Figure 5, Figure 6, Figure 7, Figure 8, Figure 9, Figure 10 and Figure 11). Figure 1 shows Figure 2, Figure 3, Figure 4, Figure 5, Figure 6, Figure 7, Figure 8, Figure 9, Figure 10 and Figure 11 consist of three images: (A) anatomical cross-section, (B) lung CT window, and (C) bone CT window. The images are arranged in a rostrocaudal sequence from the beak to the orbital fossa.

### 3.1. Anatomical Sections

We identified clinically relevant structures of the nasal cavity by anatomical cross-sections. Therefore, the slices chosen displayed the important structures of this cavity that extends from the nostrils to the choanal slit. As in other birds, the nostrils were located at the base of the beak, which could be distinguished in the sagittal images (Figure 10A). The transversal and dorsal images allowed for the observation of the nasal septum, which separates the left and right sides of the nasal cavity (subfigure A in Figure 2, Figure 3, Figure 4, Figure 5, Figure 6, Figure 7, Figure 8, Figure 9 and Figure 11). The floor of the nasal cavity was broadly supported in its rostral portion by the palatine processes of the maxilla. In its caudal portion, the only bones presented were the thin palatine bone and the vomer. These transversal images also facilitated the identification of the nasal cavity roof, constituted rostrally by the nasal and maxillary bones, and more caudally by the lacrimal bone (subfigure A in Figure 5, Figure 6, Figure 7, Figure 8 and Figure 9). 

In addition, these sections facilitated the observation of the cone-shaped nasal cavity, where the apex is pointed rostrally and consists caudally of three structures known as nasal conchae. The most anterior nasal concha is called the rostral nasal concha (subfigure A in Figure 3, Figure 4, Figure 10 and Figure 11), which appears as a rostrally pointing cone whose base shows a smooth and flat medial surface. Interestingly, the caudal part of the nasal concha, which corresponds to the base of the cone, is completely sealed by a flat plate of cartilage that forms the wall of the nasal cavity. It is followed by a large middle nasal concha (subfigure A in Figure 5, Figure 6, Figure 7, Figure 10 and Figure 11), which is the largest nasal concha in size height and width. More caudally, we distinguished the caudal nasal concha located rostrally to the eyeball (subfigure A in Figure 6, Figure 7, Figure 8, Figure 9, Figure 10 and Figure 11). Interestingly, this nasal concha does not connect with the nasal cavity but with the infraorbital sinus, also called the antorbital sinus. This connection was distinguished close to the angle of the eye. This sinus occupies a relatively large triangular space in the maxillary bone and extends rostroventrally to the eye, surrounded almost entirely by soft tissue (subfigure A in Figure 3, Figure 4, Figure 5, Figure 6, Figure 7, Figure 8, Figure 9 and Figure 11). In addition, we distinguished a space located between the nasal conchae and the walls of the nasal cavity, known as the nasal meatus (subfigure A in Figure 5, Figure 6, Figure 7, Figure 8, Figure 9 and Figure 11). Other structures such as the nasolacrimal duct, opening into the nasal cavity between the middle nasal conchae, were also visible (Figure 5A). This duct has a curved course, passes rostrally over the dorsal apex of the dorsal sinus of the infraorbital sinus, and then turns ventrally, joining the medial wall of the infraorbital sinus. In addition, these sections were quite helpful in depicting the infraorbital nerve, which runs along the medial wall (subfigure A in Figure 6).

These cross-sections also permitted the visualization of various formations of the oral cavity, encompassing the tongue, the oral vestibule, and the oropharynx (subfigure A in Figure 2, Figure 3, Figure 4, Figure 5, Figure 6, Figure 7, Figure 8 and Figure 9). Additionally, anatomical sections facilitated the visualization of an aperture connecting the nasal cavity to the oral cavity called the choanal cleft. This structure showed an elongated shape, with one portion being a slit and another portion being caudally triangular. The triangular section lies caudally between the palatine bone, and dorsally it is divided in the midline by the vomer and the nasal septum (subfigure A in Figure 7, Figure 8 and Figure 9). More caudal transverse sections were quite helpful in distinguishing muscle formations surrounding the choanal cleft (Figure 9A). Adjacent structures such as the nasal bone, maxillary bone, mandible, maxillary process of the palatine bone, basis cranii, parasphenoid, and lacrimal bone were also seen with the cross-sectional images (subfigure A in Figure 2, Figure 3, Figure 4, Figure 5, Figure 6, Figure 7, Figure 8, Figure 9 and Figure 10). Adjoining to the medial part of the lacrimal bone, we visualized an oval structure that corresponded to the nasal gland (Figure 8A and Figure 9A). The nasal vestibule receives secretions from this nasal gland, which moisten the entrance to the nasal cavity. Rostrally, this gland was closely related to the nasal bone medially, and the lacrimal bone to more laterally (Figure 8A). Meanwhile, a more caudal section facilitated the view of the gland in close contact with the frontal bone (Figure 9A). In addition, different muscles, including the *intermandibularis ventralis, Pterygoideus, tracheolateralis*, and *adductor mandibulae externus* muscles, were also well identified (subfigure A in Figure 2, Figure 3, Figure 4, Figure 5, Figure 6, Figure 7, Figure 8, Figure 9, Figure 10 and Figure 11). Finally, several eye structures, including the optic nerve, the extraocular muscles, the orbit, and the *camera vitrea bulbi*, were also observed (subfigure A in Figure 8, Figure 9, Figure 10 and Figure 11). Dorsal cross-sections allowed for the observation of different ear structures, such as the external acoustic meatus and the inner ear (Figure 11A).

### 3.2. Computed Tomography (CT)

CT images proved to be very useful in identifying the key structures that make up the nasal cavity. Therefore, the pulmonary and bone CT windows provided accurate details of the nasal septum, revealing a linear shape and displaying hypoattenuation with relation to adjacent bones (subfigures B, C in Figure 1, Figure 2, Figure 3, Figure 4, Figure 5, Figure 6, Figure 7, Figure 8, Figure 9 and Figure 11). Moreover, these two windows were essential in distinguishing the different conchae. Thus, we visualized the rostral, medial, and caudal conchae (subfigures B, C in Figure 2, Figure 3, Figure 4, Figure 5, Figure 6, Figure 7, Figure 8, Figure 9, Figure 10 and Figure 11). In addition to observing the different nasal conchae, we identify hypoattenuating passages or spaces between the conchae, which correspond to the nasal meatus (subfigures B, C in Figure 5, Figure 6, Figure 7, Figure 8, Figure 9 and Figure 11).

Similarly, a hypoattenuating structure was extended from the maxillary bone and projected rostroventrally to the orbit, presenting an elongated and triangular morphology that constituted the infraorbital sinus (subfigures B, C in Figure 2, Figure 3, Figure 4, Figure 5, Figure 6, Figure 7, Figure 8, Figure 9 and Figure 11). This sinus plays an important role in the pneumatization of the skull.

The combination of transverse, sagittal, and dorsal CT images facilitated the identification of the bony formations associated with the nasal cavity. Hence, dorsal to the nasal concha, we could distinguish a hyperattenuated structure corresponding with the rostral part of the nasal bone (subfigures B, C in Figure 1, Figure 2, Figure 3, Figure 4, Figure 5, Figure 6, Figure 7, Figure 8 and Figure 10). Subsequently, we were able to distinguish a variety of bony formations, including the maxillary, frontal, basis cranii, parasphenoid rostrum, lacrimal, and palatine bones (subfigures B, C in Figure 1, Figure 2, Figure 3, Figure 4, Figure 5, Figure 6, Figure 7, Figure 8, Figure 9, Figure 10 and Figure 11). Close to the lacrimal bone and over the dorsal aspect to the eyeball, we observed an oval shape structure displaying moderate to high attenuation that corresponded with the nasal gland (Figure 8B,C and Figure 9B,C). Moreover, we also identified relevant muscles, such as the intermandibularis ventralis, Pterygoideus, tracheolateralis, and adductor mandibulae externus muscles (subfigures B, C in Figure 2, Figure 3, Figure 4, Figure 5, Figure 6, Figure 7, Figure 8 and Figure 9).

The roof of the oral cavity and pharynx was represented by the oropharynx seen in the pulmonary and bone CT windows showing a hypoattenuation related to adjacent oral structures (subfigures B, C in Figure 7, Figure 8 and Figure 9). An elongated linear hypoattenuated structure connecting the oropharynx with the left and right nasal cavities was displayed (subfigures B, C in Figure 7, Figure 8 and Figure 9). It constituted the choana, which showed a slit-like anterior part and a wider triangular posterior part. Dorsally, the choana was delineated by the surrounding bones, with the vomer and nasal septum dividing the nasal cavity at the midline (Figure 8B,C and Figure 9B,C).

Eye bulb structures were observed with moderate to intermediate density and appeared grey, including the optic nerve, the extraocular muscles, the orbit, and the vitreous chamber (subfigures B, C in Figure 8, Figure 9, Figure 10 and Figure 11). In contrast, different ear elements, such as the external acoustic meatus and the inner ear, displayed moderate to low attenuation (Figure 11B,C).

## 4. Discussion

Modern imaging techniques have revolutionized our anatomical understanding and identification of several pathologies in veterinary medicine [5,9,24,25,35,36,37,38,39,42,43,44]. Unlike traditional methods, such as radiology and ultrasound, advanced imaging technologies offer exceptional resolution of anatomical structures, precise definition of the extent and nature of lesions, rapid image acquisition, and elimination of overlapping structures. These improvements have significantly transformed veterinary research, clinical practice, and teaching [5,9,24,25,35,36,37,38,39,44]. Modern diagnostic imaging techniques have become essential tools in veterinary medicine. CT allows for detailed cross-sectional imaging of the body, which facilitates the identification of internal structures with unprecedented clarity. In addition to improvements in image quality and diagnostic accuracy, these advanced technologies have streamlined the image acquisition process. Rapid image acquisition and processing reduces anesthesia time for patients and diagnostic procedures-associated stress, also allowing for faster decision-making by veterinarians. The elimination of overlapping structures and the ability to reconstruct images in different planes have improved diagnostic clarity and image interpretation [5,9,24,25,35,36,37,38]. In education, modern imaging techniques have transformed the teaching of veterinary anatomy and pathology. The availability of detailed, three-dimensional images enables students to explore complex structures and better understand the relationship between anatomy and function [45]. Simulations based on real images facilitate more interactive and hands-on learning, better preparing future veterinarians for their professional practice.

However, these techniques in exotic animal medicine face significant limitations due to high costs, restricted availability of equipment, and the logistical challenges associated with imaging certain species. Although detailed descriptions using computed tomography (CT) and magnetic resonance imaging (MRI) of the nasal cavities and frontal sinuses exist in exotic and conventional animals, such as dogs, horses, sea turtles, rabbits, koalas, and guinea pigs, to the authors’ knowledge, no anatomical studies combining these imaging techniques have been performed in shearwaters [8,24,25,26,27,28,29,30,31,32,39]. This research represents the first description of the nasal cavity of the shearwater using multiplanar anatomical images and transverse, sagittal, and dorsal CT images.

The nasal septum separates the nasal cavity into two passages, showing a similar organization to other birds and mammals [1,2,3,4,5,6,7,8,9]. Additionally, the transverse anatomical cross-sections and CT images were essential in distinguishing the septal sinus, which contrasts with other species, where it is depicted as a single bony space [46]. Inside the nasal cavity, we could identify the nasal conchae, also known as turbinates, which were arranged in a rostrocaudal series, meaning they are aligned from the front to the back of the nasal cavity. Because of this arrangement, the terms commonly used in mammals to describe the nasal meatuses—dorsal, middle, and ventral nasal meatus—are not applicable in this context [1,2,3,4,5,6,7,8,9]. As happens in aquatic and non-aquatic bird species, including the chicken, turkey, ostrich, amazon parrot, pigeon, duck, goose, and puffin [4,5,6,7,8,9], the middle nasal concha was the largest nasal concha. Very interestingly, the CT used here was quite helpful in distinguishing the connection of the caudal nasal concha with the infraorbital sinus as described in previous reports [47,48].

In birds, the infraorbital sinus presents multiple interconnected cavities, which contrasts markedly with the simple structure of the nasal and frontal sinuses seen in many mammals [1,2,3,4,5,6,7,8,9]. These interconnected cavities involved large areas of the nasal cavity, extending from the rostral nasal concha to the ventral aspect of the eyeball. This characteristic could be confirmed in the anatomical cross-sections and the CT images. The function of this sinus remains unknown, although some reports have suggested that some birds could have the ability to move air through a portion of the sinus [10,11,12].

Advanced imaging and endoscopic techniques have already been used for the accurate visualization of these cavities in other bird species [49]. The relevant development of this sinus makes birds particularly susceptible to developing sinusitis, since it provides an environment conducive to the accumulation of exudates and the proliferation of pathogens, complicating the diagnosis and treatment of this condition. Difficulty in accessing all affected areas by conventional techniques may result in less effective treatments and a higher incidence of sinusitis recurrence [49,50,51]. In addition, anatomical variability between different avian species may require a personalized approach to the evaluation and treatment of sinus infections.

Another distinctive anatomical structure of the nasal cavity is the choana. This structure in birds and reptiles is related to the incomplete secondary palate, through which the nasal and oral cavities communicate [52]. Studies performed on these structures in different species, such as the chicken, the common kestrel, the common moorhen, and the hoopoe, reported an oval shape. Here, it showed a linear shape as demonstrated in the cross-sectional and CT images. In the clinical context, this anatomical feature offers a significant advantage for medical evaluation [51]. The choanal cleft facilitates endoscopic access to the nasal cavity, allowing for an adequate and precise analysis of the nasal cavity. In addition, this direct access through the choana allows for the efficient collection of nasal samples, which is crucial for the diagnosis and monitoring of various respiratory conditions and infectious diseases in birds [53,54].

Finally, we can report that the CT images obtained in this study had a low resolution, which could be attributed to the low tissue volume of the samples analyzed and the small dimensions of the nasal cavity (measuring around 7.5 cm from the beak to the rostral area of the orbit) rendered the acquisition of the nasal cavity images particularly challenging. This limitation in image quality could be overcome with the use of micro-computed tomography (micro-CT). Previous studies in other exotic species have shown that micro-CT can provide images with significantly higher resolution [10,55]. However, it is essential to note that micro-CT is not commonly available in veterinary hospitals, due to its complex technology and associated costs, which limit its access and routine use in veterinary practice, restricting it mainly to specialized research centres or institutions with advanced imaging resources.

Important limitations due to the age of specimens should be highlighted since all the animals studied here were juvenile. Some investigations performed on mammals reported that relevant components of the nasal cavity such as the nasal bone increase about three times in length and width when they reach puberty [51,55]. Therefore, further studies should be conducted on juvenile and adult specimens to compare the size of the different formations that compose the nasal cavity to obtain accurate morphometric data. Moreover, some of our specimens showed a clear visualization of down feathers, which helps to insulate these animals against heat loss. Among the functions of the nasal cavity are humidification of the inspired air, air filtration, water preservation, and thermoregulation [9,10,11,12,56]. These functions are shared in several bird species, including extinct species [4,57]. This information should be considered in future studies to evaluate specific morphologic findings in juvenile specimens with some of the mentioned functions.

## 5. Conclusions

This investigation represents the first detailed description of the shearwater’s nasal cavity using computed tomography (CT) images in transverse, sagittal, and dorsal planes combined with anatomical cross-sections. The pictures acquired in this manuscript were handy in offering accurate anatomical landmarks of this region. Consequently, the information obtained in this study could be of great use to clinicians in interpreting various pathological conditions of the nasal cavity. Additionally, these imaging techniques could play a crucial role in teaching applied anatomy to our residents and veterinary students, as these methods allow for the clear depiction of structures without overlapping, thus resolving the problems associated with visualizing the organization of the rostral part of the shearwater’s head. The ability to observe anatomical structures in detail and in different planes provides a deeper and more accurate understanding of the anatomy of this region. Finally, it is pivotal to note that a notable comprehension of the specific anatomy of the nasal cavity of these species, achieved through advanced imaging diagnostic techniques, lets us evaluate diseases affecting this area. This approach not only improves diagnostic accuracy but also facilitates the development of more effective and specific treatments for pathological conditions that may occur in the nasal cavity of shearwaters.

## Figures and Tables

**Figure 1 animals-14-03015-f001:**
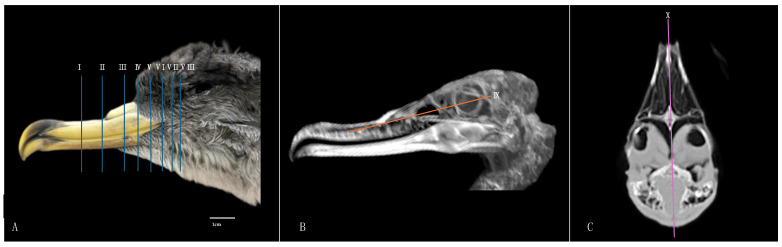
Anatomical (**A**) (labelled with blue lines), sagittal MPR (**B**) (labelled with an orange line), and horizontal (**C**) (labelled with a pink line) CT images corresponding to the approximate levels of the respective transverse, sagittal, and dorsal slices of the nasal cavity of a Cory’s Shearwater (*Calonectris borealis*).

**Figure 2 animals-14-03015-f002:**
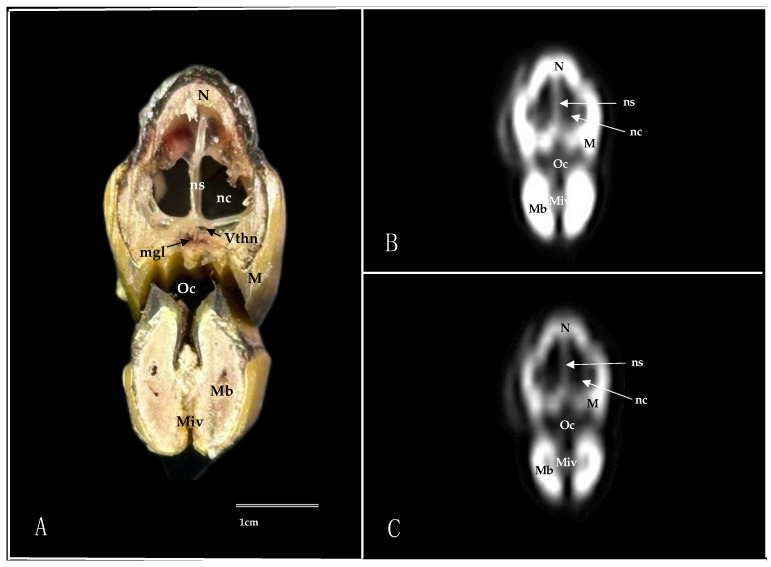
Transverse cross-section (**A**), pulmonary window (**B**), and bone window(**C**) CT images of the Cory’s Shearwater’s nasal cavity at the level of the beak, corresponding to line I in Figure 1. M: maxillary bone; Mb: mandible; mgl: maxillary salivary gland; Miv: *Musculus intermandibularis ventralis*; N: nasal bone; nc: nasal cavity; ns: nasal septum; Oc: oral cavity; Vthn: Ophthalmic division of trigeminal nerve.

**Figure 3 animals-14-03015-f003:**
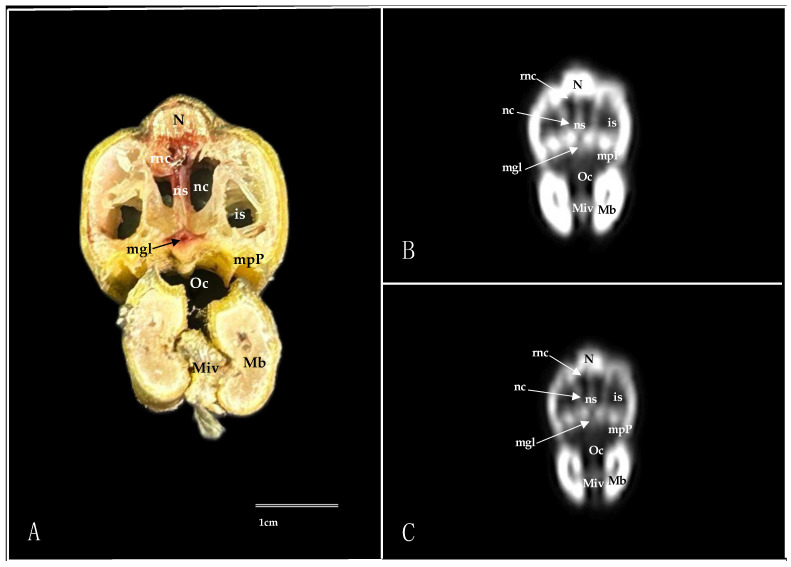
Transverse cross-section (**A**), pulmonary window (**B**), and bone window (**C**) CT images of the Cory’s Shearwater’s nasal cavity at the level of the rostral nasal concha, corresponding to line II in Figure 1. is: infraorbital sinus; Mb: mandible; mgl: maxillary salivary gland; Miv: *Musculus intermandibularis ventralis*; mpP: maxillary process of palatine bone; N: nasal bone; nc: nasal cavity; ns: nasal septum; Oc: oral cavity; rnc: rostral nasal concha.

**Figure 4 animals-14-03015-f004:**
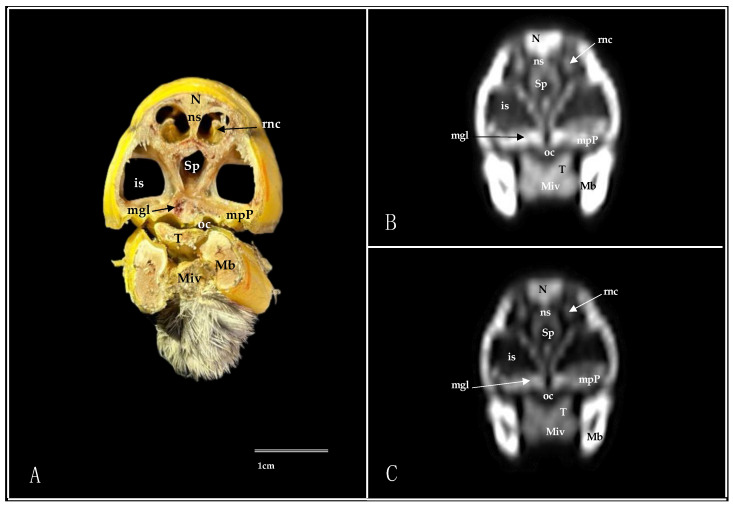
Transverse cross-section (**A**), pulmonary window (**B**), and bone window (**C**) CT images of the Cory’s Shearwater’s nasal cavity at the level of the Maxillary salivary gland, corresponding to line III in Figure 1. is: infraorbital sinus; Mb: mandible; mgl: maxillary salivary gland; Miv: *Musculus intermandibularis ventralis*; mpP: maxillary process of palatine bone; N: nasal bone; ns: nasal septum; Oc: oral cavity; rnc: rostral nasal concha; Sp: *Sinus septalis*; T: tongue.

**Figure 5 animals-14-03015-f005:**
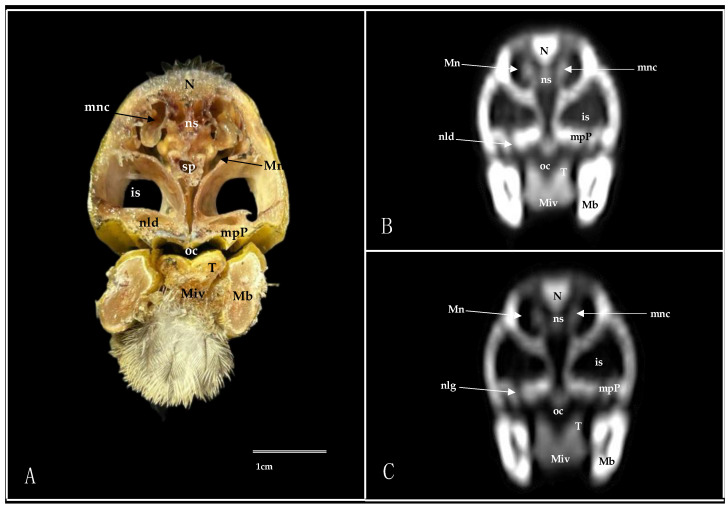
Transverse cross-section (**A**), pulmonary window (**B**), and bone window (**C**) CT images of the Cory’s Shearwater’s nasal cavity at the level of the nasal gland, corresponding to line IV in Figure 1. is: infraorbital sinus; Mb: mandible; Miv: *Musculus intermandibularis ventralis*; Mn: *Meatus nasalis*; mnc: middle nasal concha; mpP: maxillary process of palatine bone; N: nasal bone; nld: nasolacrimal duct; ns: nasal septum; oc: oral cavity; Sp: *Sinus septalis*; T: tongue.

**Figure 6 animals-14-03015-f006:**
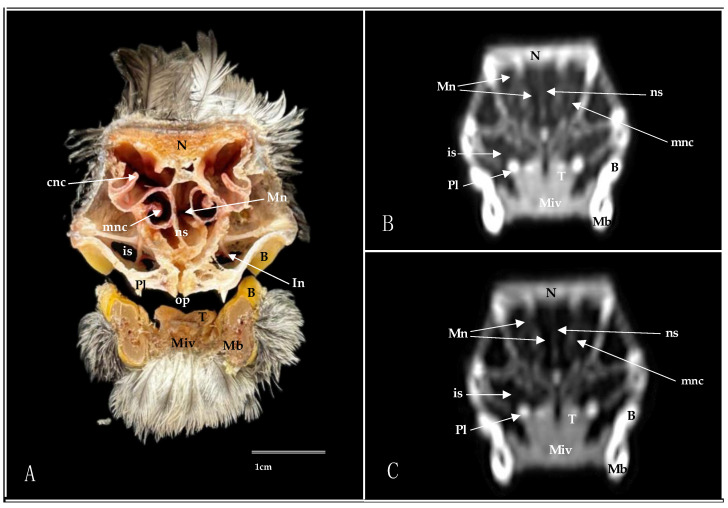
Transverse cross-section (**A**), pulmonary window (**B**), and bone window (**C**) CT images of the Cory’s Shearwater’s nasal cavity at the level of the infraorbital nerve, corresponding to line V in Figure 1. B: beak; cnc: caudal nasal concha; In: infraorbital nerve; is: infraorbital sinus; Mb: mandible; Miv: *Musculus intermandibularis ventralis*. Mn: *Meatus nasalis*; mnc: middle nasal concha; N: nasal bone; ns: nasal septum; op: oropharynx; Pl: palatine bone; T: tongue.

**Figure 7 animals-14-03015-f007:**
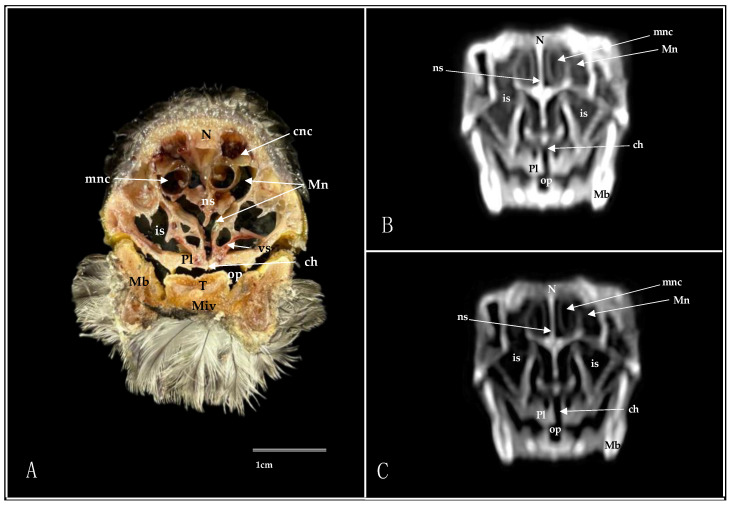
Transverse cross-section (**A**), pulmonary window (**B**), and bone window (**C**) CT images of the Cory’s Shearwater’s nasal cavity at the level of the medial nasal concha, corresponding to line VI in Figure 1. ch: choanal cleft; cnc: caudal nasal concha; is: infraorbital sinus; Mb: mandible; Miv: *Musculus intermandibularis ventralis*; mnc: middle nasal concha; Mn: *Meatus nasalis*; N: nasal bone; ns: nasal septum; op: oropharynx; Pl: palatine bone; T: tongue; vs: vascular plexus.

**Figure 8 animals-14-03015-f008:**
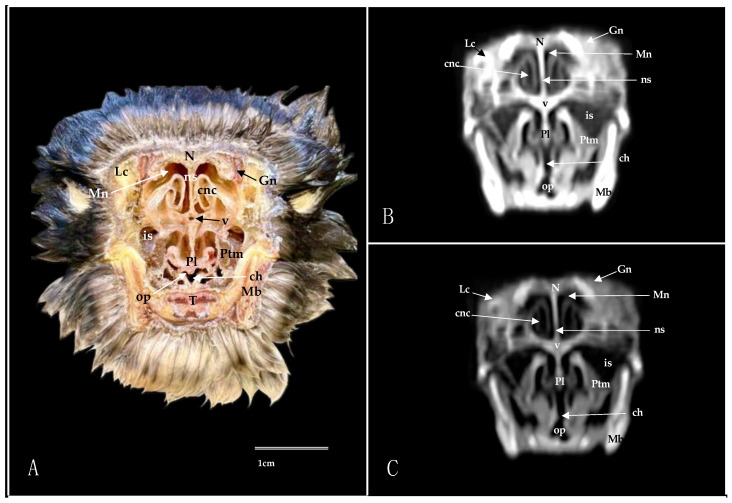
Transverse cross-section (**A**), pulmonary window (**B**), and bone window (**C**) CT images of the Cory’s Shearwater’s nasal cavity at the level of the caudal nasal concha, corresponding to line VII in Figure 1. ch: choanal cleft; cnc: caudal nasal concha; Gn: *Glandula nasalis*; is: infraorbital sinus; Lc: lacrimal bone; Mb: mandible; Mn: *Meatus nasalis*; N: nasal bone; ns: nasal septum; op: oropharynx; pl: palatine bone; Ptm: pterygoid muscle; T: tongue; v: vomer.

**Figure 9 animals-14-03015-f009:**
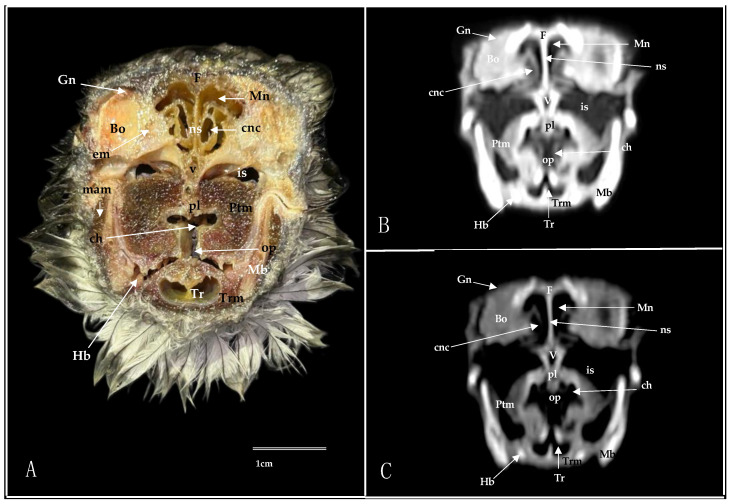
Transverse cross-section (**A**), pulmonary window (**B**), and bone window (**C**) CT images of the Cory’s Shearwater’s nasal cavity at the level of the *bulbus oculi*, corresponding to line VIII in Figure 1. Bo: *bulbus oculi*; ch: choanal cleft; cnc: caudal nasal concha; em: extraocular muscle; F: frontal bone; Gn: *Glandula nasalis*; Hb: hyobranchial apparatus; is: infraorbital sinus; mam: *Musculus adductor mandibulae externus*; Mb: mandible; Mn: *Meatus nasalis*; ns: nasal septum; op: oropharynx; pl: palatine bone; Ptm: pterygoide muscle; Tr: trachea; Trm: tracheolateralis muscle; v; vomer.

**Figure 10 animals-14-03015-f010:**
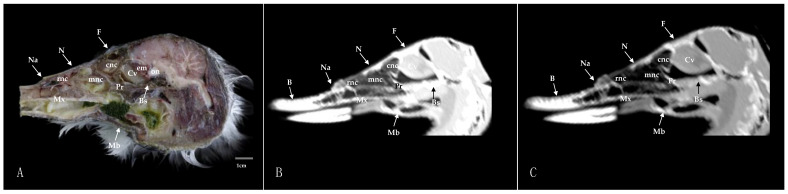
Sagittal cross-section (**A**), pulmonary window (**B**), and bone window (**C**) CT images of the nasal cavity of the Cory’s Shearwater’s. B: beak; Bs: *basis cranii*; cnc; caudal nasal concha; Cv: *camera vitrea bulbi*; em: extraocular muscle; F: frontal bone; Mb: mandible; mnc: middle nasal concha; Mx: maxilla; N: nasal bone; Na: nares; on: optic nerve; Pr: parasphenoid rostrum; rnc: rostral nasal concha.

**Figure 11 animals-14-03015-f011:**
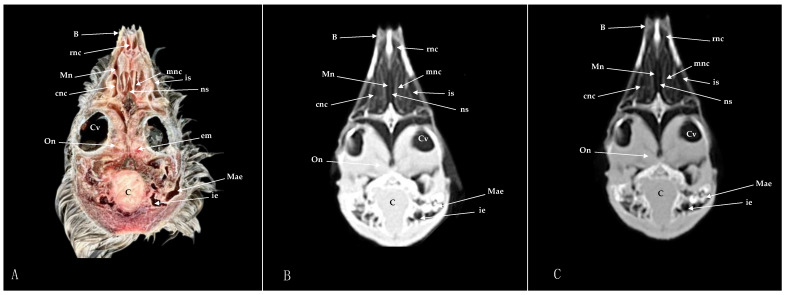
Dorsal cross-section (**A**), pulmonary window (**B**), and a mix of bone/pulmonary and soft-tissue algorithm (**C**). CT images of the nasal cavity of the Cory’s Shearwater at the level of the nares corresponding to line IX in Figure 1. B: beak; C: *Cerebellum* (body); cnc: caudal nasal concha; Cv: *camera vitrea bulbi*; em: extraocular muscle; ie: inner ear; Mae: *Meatus acusticus externus*; Mn: nasal meatus; mnc: medial nasal concha; ns: nasal septum; On: ocular nerve; rnc: rostral nasal concha.

## Data Availability

The data supporting the reported results can be found at https://accedacris.ulpgc.es/ (accessed on 10 October 2024).

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
