# Peer review of "Computed Tomography Anatomy of the Juvenile Cory’s Shearwater (Calonectris borealis) Normal Nasal Cavity"

_animals, 2024, doi:10.3390/ani14203015_

Round 1

Reviewer 1 Report

Comments and Suggestions for Authors

I congratulate the authors on this more focused, comparative exploration of shearwater anatomy. I found it very useful and informative. The authors laid out their aim, offered an appropriate level of detail with their methodology, and provided several clinically-relevant examples of how this approach could help exotic vets in their practice.

Aside from these small suggestions I believe that the paper is suitable for publication.

Minor corrections
___________________

The authors use infraorbital sinus, which is often seen in the veterinary literature. However, it is also referred to as the antorbital sinus by other researchers. I would toss in a quick mention of this when the infraorbital sinus is first discussed. For example: "...infraorbital sinus (= antorbital sinus sensu Witmer 1995)." This will let future readers know about this synonymy should they want to delve deeper into this part of avian anatomy.

Line 46: The sentence: "This part consists mainly of a cartilaginous structure" can be deleted as it just restates what the sentence just before it said.

Line 147:  "Interestingly, the caudal nasal concha does not connect with the nasal cavity but with the infraorbital sinus." -- This is the only time this connection is mentioned, and it would seem to be in contrast with what we know about the avian nasal passages from other researchers. The caudal concha houses olfactory epithelium and should only function when inhaled air passes over that epithelium. If the concha connects directly to the infraorbital sinus instead of the nasal passage, then that wouldn't happen. I would urge the authors to re-evaluate this connection.

Lines: 165 & 192: Parasfenoid should be: parasphenoid. The former seems to be an older, German or Czech spelling of the Greek words.

Line 185: "This sinus plays an important role in the respiratory physiology of birds, facilitating ventilation and contributing to the cooling system of the air before it reaches the lungs." -- My understanding of these paranasal sinuses is that their function (if any) remains unknown. The sinuses are mostly dead-end spaces with little airflow, but some birds may have a minor ability to move a bit of air through portions of the sinus. If the authors could offer a citation here, I think it would help.

Lastly, since this is the avian nasal passage, I would encourage the authors to cite Betsy Bang's influential works on the matter, especially when discussing the comparative anatomy.

References

Bang, B. G. 1971. Functional anatomy of the olfactory system in 23 orders of birds. Acta Anatomica, 79(58), 176.

Witmer, L. M. 1995. Homology of facial structures in extant archosaurs (birds and crocodilians), with special reference to paranasal pneumaticity and nasal conchae. Journal of Morphology, 225, 269327.

Author Response

Dear Reviewer,

We really appreciate your comments about the manuscript. Moreover, the minor corrections have been quite helpful in improving our manuscript and helped to reach a broader vision.

Comment 1:

The authors use infraorbital sinus, which is often seen in the veterinary literature. However, it is also referred to as the antorbital sinus by other researchers. I would toss in a quick mention of this when the infraorbital sinus is first discussed. For example: "...infraorbital sinus (= antorbital sinus sensu Witmer 1995)." This will let future readers know about this synonymy should they want to delve deeper into this part of avian anatomy.

As you recommend, we have included antorbital sinus in the results section as it is in the first time is mentioned.

Comment 2:

Line 46: The sentence: "This part consists mainly of a cartilaginous structure" can be deleted as it just restates what the sentence just before it said.

Following this recommendation, we have deleted this sentence.

Commment 3:

Line 147:  "Interestingly, the caudal nasal concha does not connect with the nasal cavity but with the infraorbital sinus." -- This is the only time this connection is mentioned, and it would seem to be in contrast with what we know about the avian nasal passages from other researchers. The caudal concha houses olfactory epithelium and should only function when inhaled air passes over that epithelium. If the concha connects directly to the infraorbital sinus instead of the nasal passage, then that wouldn't happen. I would urge the authors to re-evaluate this connection.

Following your recommendation, we have reevaluated this connection. Therefore, we followed the caudal course of the infraorbital sinus, where we distinguished its communication with the caudal nasal concha. 

Comment 4:

Lines: 165 & 192: Parasfenoid should be: parasphenoid. The former seems to be an older, German or Czech spelling of the Greek words.

As you recommended, we have replaced "parasfenoid" by "parasphenoid".

Comment 5:

Line 185: "This sinus plays an important role in the respiratory physiology of birds, facilitating ventilation and contributing to the cooling system of the air before it reaches the lungs." -- My understanding of these paranasal sinuses is that their function (if any) remains unknown. My understanding of these paranasal sinrrruses is that their function (if any) remains unknown. The sinuses are mostly dead-end spaces with little airflow, but some birds may have a minor ability to move a bit of air through portions of the sinus.My understanding of these paranasal sinuses is that their function (if any) remains unknown. The sinuses are mostly dead-end spaces with little airflow, but some birds may have a minor ability to move a bit of air through portions of the sinus. If the authors could offer a citation here, I think it would help.

We have considered  your explanation and deleted this paragrah.

Comment 6:

Following your recommendation, we have added various works in the reference list, including those of Betsy Bang.

Reviewer 2 Report

Comments and Suggestions for Authors

In this manuscript the authors present the gross sectional anatomy of the nasal cavity of juvenile Cory's shearwater (Calonectris borealis). This study follows a previously published paper (doi: 10.3390/ani14131962). Although this work might be interesting to researchers in the field of ornithology, some issues should be addressed prior to publication:

- Detailed anatomy of the avian nasal cavity should be described.

- Since this is a post mortem study, histological imaging might help identifying anatomical structures.

- Please provide detailed morphometrical data of the nasal cavity.

- 3D-reconstruction might help the reader understanding the topographical anatomy.

- Since juvenile birds were used in this study, the influence of age should be discussed.

- Please discuss the physiological and phylogenetic consequences of the morphological findings.

Comments on the Quality of English Language

Minor spelling errors.

Author Response

Dear Reviewer 2,

We have considered all your comments to improve the quality of our manuscript. This manuscript is a continuation of the article that you already mentioned, providing a deeper and specific analysis of the nasal cavity and paranasal sinuses.

Comment 1:

Detailed anatomy of the avian nasal cavity should be described.

Following your recommendation, we have added additional information about the avian nasal cavity in the results section (highlighted in red).

Comment 2:

Since this is a post-mortem study, histological imaging might help identify anatomical structures. 

We agree with this comment. Nonetheless, we focused only on the anatomical cross-sections and their correlation with the CT images to better understand the normal anatomy, facilitating their use to veterinary practitioners, radiologists and researchers. Thus, this combination (CT images in different planes and windows) provided essential insights into anatomical identification, as mentioned by reviewer 1.

Comment 3.

Please provide detailed morphometrical data of the nasal cavity.

Thank you for pointing this out. However, it is important to highlight that our study was done in specimens with small body sizes compared to other studies that performed morphometric analysis in larger species, including turkeys and ostrichs. Interestingly, these studies reported the physical limitations of these measurements even as body size increases.

  1. Bourke, J.M.; Witmer, L.M. Soft Tissues Influence Nasal Airflow in Diapsids: Implications for Dinosaurs. J Morphol 2023, 284, e21619, doi:10.1002/JMOR.21619
  2. Madkour, F.A. Beak, Oropharyngeal and Nasal Cavities of Broad Breasted White Turkey (Meleagris Gallopavo): Gross Anatomical and Morphometrical Study. J Adv Vet Res 2022, 12, 99–106

This point is essential since here we should remember the small size of the nasal cavity in these animals (with the largest sections less than 1x1 cm) and the age of specimens, which produce important limitations in providing accurate morphometric data.

Comment 4:

 3D-reconstruction might help the reader understanding the topographical anatomy.

As you suggest, we performed 3D reconstructions on our specimens. However, the reconstruction quality was quite low and due to this, we decided to exclude these images in the manuscript. It was due to the small size and tisular volume of these specimens. With the use  of micro-CT equipment, we could obtain these specific information. Unfortunately, these types of equipment are not included in conventional veterinary hospitals.

Comment 5:

Since juvenile birds were used in this study, the influence of age should be discussed.

Following your recommendation, we have added a paragragh, where we discuss the age influence supported by specific references.

Comment 6:

Please discuss the physiological and phylogenetic consequences of the morphological findings.

We appreciate this interesting comment, therefore, we have discussed these specific concerns related to the morphological findings.

Round 2

Reviewer 2 Report

Comments and Suggestions for Authors

All issues have been addressed in a sufficient way.

Author Response

Dear Reviewer,

We really apreciate your help in order to improve the quality of our manuscript. Moreover, all these comments were essential to clarify the limitations identified in this research.